# Network pharmacology analysis and molecular docking to unveil the potential mechanisms of San-Huang-Chai-Zhu formula treating cholestasis

**Binbin Liu, Jie Zhang, Lu Shao, Jiaming Yao**[ID]*

Department of Digestion, Hangzhou TCM Hospital Affiliated to Zhejiang Chinese Medical University (Hangzhou Hospital of Traditional Chinese Medicine), Hangzhou, Zhejiang, China

* turtle82@126.com

**Data Availability Statement:** All relevant data are within the paper and its Supporting Information files.

## Abstract

### Objective

Chinese medicine formulae possess the potential for cholestasis treatment. This study aimed to explore the underlying mechanisms of San-Huang-Chai-Zhu formula (SHCZF) against cholestasis.

### Methods

The major chemical compounds of SHCZF were identified by high-performance liquid chromatography. The bioactive compounds and targets of SHCZF, and cholestasis-related targets were obtained from public databases. Intersected targets of SHCZF and cholestasis were visualized by Venn diagram. The protein-protein interaction and compound-target networks were established by Cytoscape according to the STRING database. The biological functions and pathways of potential targets were characterized by Gene Ontology and Kyoto Encyclopedia of Genes and Genomes enrichment analysis. The biological process-target-pathway network was constructed by Cytoscape. Finally, the interactions between biological compounds and hub target proteins were validated via molecular docking.

### Results

There 7 major chemical compounds in SHCZF. A total of 141 bioactive compounds and 83 potential targets were screened for SHCZF against cholestasis. The process of SHCZF against cholestasis was mainly involved in AGE-RAGE signaling pathway in diabetic complications, fluid shear stress and atherosclerosis, and drug metabolism-cytochrome P450. ALB, IL6, AKT1, TP53, TNF, MAPK3, APOE, IL1B, PPARG, and PPARA were the top 10 hub targets. Molecular docking showed that bioactive compounds of SHCZF had a good binding affinity with hub targets.

**Funding:** This work was supported by Zhejiang science and technology research fund [No. 2014C33238] and Zhejiang science and technology research fund of traditional Chinese medicine [No.2020ZB158], funder play the role of conception and design of the research, analysis and interpretation of data and revision of manuscript.

**Competing interests:** The authors have declared that no competing interests exist.

## Conclusions

This study predicted that the mechanisms of SHCZF against cholestasis mainly involved in AGE-RAGE signaling pathway in diabetic complications, fluid shear stress and atherosclerosis, and drug metabolism-cytochrome P450. Moreover, APOE, AKT1, and TP53 were the critical hub targets for bioactive compounds of SHCZF.

## Introduction

Cholestasis is a common clinical manifestation of liver disease mainly derived from the reduction or obstruction of bile flow [1]. The long-term cholestasis in liver can lead to hepatocyte dysfunctions, thereby causing severe liver diseases such as primary biliary cirrhosis, primary sclerosing cholangitis and secondary sclerosing cholangitis [2]. At present, although some drugs, such as rosiglitazone, obeticholic acid, and ursodeoxycholic acid, have been developed for cholestasis treatment, the therapeutic effect is still limited and may contribute to pruritus, dyslipidemia, and gastrointestinal symptoms [3, 4]. Therefore, the discovery of effective drugs for cholestasis treatment is of great significance.

Accumulating evidence indicated that Chinese medicines exert beneficial therapeutic effects in liver diseases and cholestasis [5, 6]. San-Huang-Chai-Zhu formula (SHCZF) is a Chinese herbal formula, which consists of five herbs, namely, Dahuang (*Rhei Radix Et Rhizome*), Huangbai (*Phellodendri Chinrnsis Cortex*), Huangzhizi (*Gardeniae Fructus*), Chaihu (*Radix Bupleuri*), and Baizhu (*Atractylodes Macrocephala Koidz.*). Previous studies indicated that these five herbs all possess the hepatoprotective effect on liver diseases. Cao et al. [7] reported that Dahuang had extensive pharmacological effects in hepatoprotective, anti-inflammatory, anticancer and so on. Huangbai and Huangzhizi were widely used to ameliorate inflammation and hepatotoxicity as a core component of herbal formula [8, 9]. Saikosaponins extracted from Chaihu showed valuable pharmacological activities of anti-inflammatory and liver protection [10]. Baizhu in Xiaoyao San formula was also validated its pharmacological effects of hepatoprotection [11]. However, the underlying pharmacological mechanism of SHCZF against cholestasis is still illusive.

Network pharmacology is a favorable method to reveal the pharmacological mechanism of Chinese medicine formulae against specific diseases and identify the relevant drugs, targets, and pathways [12–14]. This approach comprehensively investigates the interactions of bioactive ingredients, targets, and diseases, and the relationship are visualized by interaction networks. For instance, by combining the network pharmacology with the pathological examination, Xiaoyaosan decoction was proved the therapeutic effects on alleviating liver fibrosis [15]. The potential biological mechanisms of GegenQinlian decoction also were unveiled to improve insulin resistance in liver, adipose, and muscle tissue by network pharmacology analysis [16]. Therefore, network pharmacology is a commendable approach for exploring the underlying mechanisms of SHCZF against cholestasis.

In this article, the underlying mechanisms of SHCZF against cholestasis were uncovered by identifying bioactive compounds and potential target genes. Moreover, the interactions between major bioactive compounds and hub target proteins were validated by molecular docking. This study provides an essential foundation for further experimental investigations and clinical application of SHCZF against cholestasis.

## Methods

### Main ingredients analysis of SHCZF

SHCZF was prepared by mixing five herbs (Dahuang, Huangbai, Huangzhizi, Chaihu, and Baizhu) in the ratio of 4:4:3:3:4. The extract of SHCZF was obtained by adding 10 times the amount of water, soaking for 30 min, and boiling for 1.5 h. After filtering out the liquid, samples were added 8 times the amount of water and decocted for 0.5 h after boiling. Then, the obtained extract was concentrated into 2 g/mL for high-performance liquid chromatography (HPLC) determination. Samples were analyzed using a LC-20AT HPLC system (Shimadzu, Japan) and separated using an Extend-C18 column (250 mm × 4.6 mm, 5 μm) (Agilent, CA, USA) with a mobile phase consisting of 0.1% phosphoric acid (A) and acetonitrile (B). The elution gradient was as follows: 0–10 min with 90% A and 10% B, 10–20 min with 30% A and 30% B, 20–30 min with 40% A and 60% B, 30–53 min with 30% A and 70% B, 53–54 min with 90% A and 10% B, and 59 min controller stop. The molecular structures of these seven compounds of SHCZF were downloaded from ZINC (https://zinc15.docking.org/) [17].

### Screening for bioactive ingredients and targets of SHCZF

All ingredients from 5 herbs of SHCZF were retrieved from the traditional Chinese medicine integrated database (TCMID, http://www.megabionet.org/tcmid/) [18], the traditional Chinese medicine systems pharmacology database and analysis platform (TCMSP, https://old.tcmsp-e.com/tcmsp.php) [19], and herb ingredients' targets (HIT, http://lifecenter.sgst.cn/hit/) database [20]. Totally, 227 compounds were obtained after eliminating those compounds without targets. In addition, Search tool for interacting chemicals (STITCH, http://stitch.embl.de/) database [21] and the above data sources were used to retrieve targets associated with 227 compounds from SHCZF with a setting of minimum required interaction score = 0.400 in STITCH. A total of 5216 targets was collected and the Gene ID of these targets was normalized by National Center for Biotechnology Information (NCBI) database (https://www.ncbi.nlm.nih.gov/).

### Drug-likeness calculation of SHCZF compounds

The 227 compounds of SHCZF were screened by drug-likeness evaluation. The assessment of drug-likeness properties is mainly determined by absorption, distribution, metabolism, and elimination (ADME) features of compounds [22]. The quantitative estimate of drug-likeness (*QED*) value is an important parameter to assess ADME characteristics. In this work, we calculated *QED* value described by Bickerton [23] to screen pharmaceutically active compounds in SHCZF. The equation of *QED* calculation was shown as follows:

$$QED = \exp\left(\frac{1}{n}\sum_{i=1}^{n} \ln d_i\right)$$

In this equation, desirability functions (*d*) were obtained by integrating 8 physicochemical properties of compounds, including molecular weight (MW), the number of hydrogen bond acceptors (HBAs), the number of hydrogen bond donors (HBDs), the octanol-water partition coefficient (ALogP), the number of rotatable bonds (ROTBs), the number of aromatic rings (AROMs), molecular polar surface area (PSA), and the number of structural alerts (ALERTS). Compounds in SHCZF with *QED* ≥ 0.2 referring to the DrugBank database (https://go.drugbank.com/) were included for following analyses.

## Target selection of active compounds in SHCZF

To precisely define compound-target interaction, the enrichment scoring algorithm based on a binomial statistical model was used to screen core targets of compounds [24, 25]. The target that interacts with most of active compounds can be considered as a core target of SHCZF. The probability of being a core target was calculated as follows:

$$P_i(X \geq k) = \sum_{m=k}^{n} C_n^m (p)^m (1-p)^{n-m}$$

where, $n$ is the total number of compounds in SHCZF, $p$ is the ratio of the average number of compounds simultaneously interacting with the same target in the total target of SHCZF compounds, and $P_i (X \geq k)$ represents the probability of a target gene ($i$) simultaneously interacting with more than $k$ active compounds. The investigated target with $P < 0.01$ can be regarded as a core target for SHCZF compounds.

## Screening of targets associated with cholestasis

The cholestasis-related targets were retrieved from the GeneCards database (https://www.genecards.org/) [26, 27], the online mendelian inheritance in man (OMIM, https://www.omim.org/) database [28], and the DisGeNET database (https://www.disgenet.org/home/) [29]. Accordingly, 56, 28, and 420 cholestasis-related targets were collected from GeneCards, OMIM, and DisGeNET databases, respectively. A total of 449 targets was obtained after removing duplicates (S1 Table).

## Construction of protein-protein interaction (PPI) network and compound-target (C-T) network

The intersection targets of SHCZF and cholestasis were visualized by a Venn diagram. PPI network of common target proteins was established and analyzed using Search Tool for the Retrieval of Interacting Genes/Proteins (STRING) dataset (https://string-db.org) [30], where each node in the network represented a target, and the node with higher degree means the more important target in the network. The C-T network of SHCZF against cholestasis was constructed using Cytoscape (v3.8.2) [31].

## Biological function enrichment analyses

In order to further explore the biological functions of SHCZF acting on cholestasis, core targets were integrated for Gene Ontology (GO) and Kyoto Encyclopedia of Genes and Genomes (KEGG) enrichment analyses [32]. GO enrichment analysis included molecular function (MF), biological process (BP), and cellular component (CC) analyses according to the GO database. KEGG enrichment analysis were performed according to the KEGG database. A hypergeometric distribution model was used to assess whether the core target genes were significantly related to specific GO terms and KEGG pathways [33], showed as follows:

$$P = 1 - \sum_{i=0}^{k-1} \frac{\binom{M}{i}\binom{N-M}{n-i}}{\binom{N}{n}}$$

where, $N$ is the total number of genes, $M$ is the number of genes annotated in GO and KEGG databases, $n$ is the number of investigated target genes of SHCZF, and $k$ is the number of

intersection genes of SHCZF and annotated genes. *P*-values that corrected by the Bonferroni method reflected the relevance between potential targets and GO terms or KEGG pathways. GO terms and KEGG pathways with *P*-value < 0.01 were considered as significant relevance. Bubble charts and histograms were drawn based on the cluster profiler package R 3.15.4.

## Construction of a target-pathway network for SHCZF against cholestasis

To elucidate the pharmacological mechanism of SHCZF in cholestasis treatment, Cytoscape was used to construct a BP-target-pathway network. The degree, betweenness and centeredness of potential target were calculated by a CytoHubba plugin [34]. The core targets, top 15 KEGG pathways, and top 15 BPs were included in the network. Targets with flesh-colored circles, pathways with green circles, and BPs with purple circles were presented as nodes, and the interactions between nodes were expressed as edges.

## Molecular docking

Molecular docking was conducted to validate the interactions between bioactive compounds and target proteins of SHCZF against cholestasis. The top 10 hub target proteins were selected for molecular docking and used for PPI network construction by a CytoHubba plugin in Cytoscape. The 3D structures of target proteins were obtained from Protein Data Bank (PDB, https://www.rcsb.org/) [35]. After deleting water molecules using PyMol (v2.3.0) [36], the obtained protein structures were imported into AutoDockTools (v1.5.6) to construct mating pocket of molecular docking. Molecular docking with bioactive compounds was performed using AutoDock Vina (v1.1.2) [37] based on the data collected above.

# Results

## Major ingredients in SHCZF

HPLC was performed to identify the major chemical compounds in SHCZF. Seven main compounds of SHCZF were obtained, including chrysophanol, emodin, physcion, rhein, aloe-emodin, berberine chloride, gardenoside (S1A–S1G Fig). The chemical structures of these 7 compounds were shown in Table 1.

## Bioactive components and targets of SHCZF

*QED* is a critical indicator to evaluate the drug-likeness of compounds. According to the *QED* values, 216 drug-likeness components in SHCZF were obtained based on the TCMID, TCMSP, and HIT database. Moreover, 162 active compounds and 457 SHCZF compound-related targets were collected by combining the public databases with a binomial statistical

**Table 1. Chemical structures of 7 major compounds of San-Huang-Chai-Zhu formula (SHCZF).**

| Synonyms | Cas | Molecular Formula |
|---|---|---|
| Chrysophanol | 481-74-3 | $C_{15}H_{10}O_4$ |
| Emodin | 518-82-1 | $C_{15}H_{10}O_5$ |
| Physcion | 521-61-9 | $C_{16}H_{12}O_5$ |
| Rhein | 478-43-3 | $C_{15}H_8O_6$ |
| Aloe-emodin | 481-72-1 | $C_{15}H_{10}O_5$ |
| Berberine chloride | 633-65-8 | $C_{20}H_{18}ClNO_4$ |
| Gardenoside | 24512-62-7 | $C_{17}H_{24}O_{11}$ |

model. There were 19, 40, 34, 93, and 22 bioactive compounds in Dahuang, Huangbai, Huangzhizi, Chaihu, and Baizhu of SHCZF, respectively (S2 Table).

## Potential targets of SHCZF active compounds for cholestasis treatment

According to the GeneCards, OMIM, and DisGeNET databases, a total of 449 cholestasis-related target genes were obtained after eliminating duplicates (S1 Table). The intersection between 457 SHCZF targets and 449 cholestasis-related targets was presented by a Venn diagram. As a result, there were 83 overlapping targets considered as core targets associated with both SHCZF compounds and cholestasis (Fig 1A & Table 2). Furthermore, 83 potential targets were input into the STRING database to construct a PPI network. Nodes and edges in the PPI network represent targets and protein-protein associations, respectively. The PPI network included 83 nodes and 1034 edges. Green and yellow circles in the PPI network stood for 83 potential targets. The degree of targets represents the number of links to nodes, and the target with higher degree can be regarded as the more important target. In this PPI network, the darker green circles mean the targets with higher degree and yellow circles mean less importance. The average node degree of this PPI network was 24.9, and ALB, IL6, AKT1, TP53, TNF, MAPK3, APOE, IL1B, PPARG, and PPARA were top 10 targets with high degrees (Fig 1B).

## Compound-target (C-T) network of SHCZF against cholestasis

According to 83 potential targets, 141 SHCZF compounds were identified as the major ingredients acting on cholestasis (Table 3). The interactions between 83 potential targets and 141 SHCZF compounds were exhibited by a C-T network. In the C-T network, red diamonds represented 5 herbs in SHCZF, including Dahuang (*Rhei Radix Et Rhizome*), Huangzhizi (*Gardeniae Fructus*), Baizhu (*Atractylodes Macrocephala Koidz.*), Huangbai (*Phellodendri Chinrnsis Cortex*), and Chaihu (*Radix Bupleuri*). Circles with 5 different colors stood for distinct compounds from 5 herbs, among which, there were 17 compounds from *Rhei Radix Et Rhizome*, 19 from *Gardeniae Fructus*, 17 from *Atractylodes Macrocephala Koidz.*, 15 from *Phellodendri Chinrnsis Cortex*, and 49 from *Radix Bupleuri*. Besides, 24 common compounds were displayed using blue circles. The parallelograms in the network represented 83 potential targets of SHCZF against cholestasis and darker orange indicated higher degree (Fig 2).

## GO and KEGG enrichment analyses

To elaborate the biological functions of 83 potential targets, targets were characterized by GO and KEGG pathway enrichment analyses. In the GO analysis, a total of 1617 GO terms were found, including 91 of MF, 1498 of BP, and 28 of CC ($p$ value < 0.01). The top 15 terms of MF, BP, and CC were ranked according to the adjusted $p$ value and gene count (Fig 3). Lower $p$ value with red color and higher count with bigger circle indicated greater enrichment of GO terms. The bubble chart and histogram showed that MF was significantly enriched in heme binding, tetrapyrrole binding, carboxylic acid binding, receptor agonist activity, and organic acid binding, etc. (Fig 3A and 3B). The main GO terms of BP were related to response to lipopolysaccharide, regulation of lipid localization, cellular response to biotic stimulus, regulation of inflammatory response, and response to oxidative stress, etc. (Fig 3C and 3D). CC were mainly enriched in membrane microdomain, high-density lipoprotein particle, blood microparticle, nuclear transcription factor complex, and RNA polymerase II transcription factor complex, etc. (Fig 3E and 3F).

The essential signaling pathways of SHCZF in cholestasis were displayed by KEGG pathway enrichment analysis. A total of 133 pathways were significantly associated with 83 potential

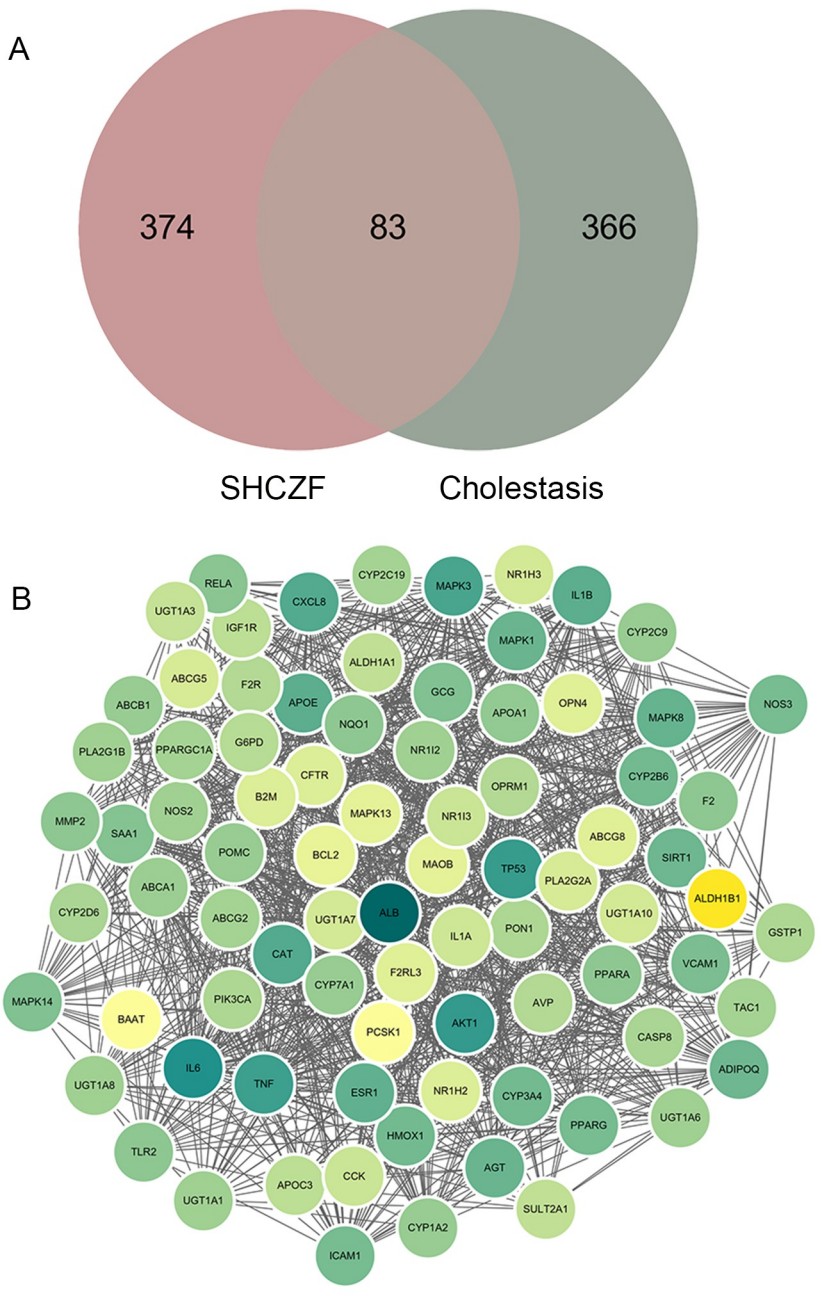

**Fig 1. The 83 potential targets for San-Huang-Chai-Zhu formula (SHCZF) in cholestasis treatment. (A)**
Intersection of SHCZF and cholestasis targets was visualized by Venn diagram. **(B)** Protein-protein interaction (PPI)
network of 83 common targets. Each node represents a common target for SHCZF and cholestasis, and each edge
represents the association between two targets. The darker green means the higher degree value, and the average
degree is 24.9.

targets ($p$ value $< 0.01$). In addition, the top 15 pathways with low adjust $p$ values and high
counts were displayed by the bubble chart and the histogram (Fig 4A and 4B), and listed in
Table 4. The results showed that the common signaling pathways mainly focused on the

**Table 2. 83 potential targets of SHCZF against cholestasis.**

| Gene ID | Target Name | Gene ID | Target Name | Gene ID | Target Name | Gene ID | Target Name |
|---------|-------------|---------|-------------|---------|-------------|---------|-------------|
| 19 | ABCA1 | 1559 | CYP2C9 | 4846 | NOS3 | 7157 | TP53 |
| 183 | AGT | 1565 | CYP2D6 | 4988 | OPRM1 | 7376 | NR1H2 |
| 207 | AKT1 | 1576 | CYP3A4 | 5122 | PCSK1 | 7412 | VCAM1 |
| 213 | ALB | 1581 | CYP7A1 | 5243 | ABCB1 | 8856 | NR1I2 |
| 216 | ALDH1A1 | 1728 | NQO1 | 5290 | PIK3CA | 9002 | F2RL3 |
| 219 | ALDH1B1 | 2099 | ESR1 | 5319 | PLA2G1B | 9370 | ADIPOQ |
| 335 | APOA1 | 2147 | F2 | 5320 | PLA2G2A | 9429 | ABCG2 |
| 345 | APOC3 | 2149 | F2R | 5443 | POMC | 9970 | NR1I3 |
| 348 | APOE | 2539 | G6PD | 5444 | PON1 | 10062 | NR1H3 |
| 551 | AVP | 2641 | GCG | 5465 | PPARA | 10891 | PPARGC1A |
| 567 | B2M | 2950 | GSTP1 | 5468 | PPARG | 23411 | SIRT1 |
| 570 | BAAT | 3162 | HMOX1 | 5594 | MAPK1 | 54575 | UGT1A10 |
| 596 | BCL2 | 3383 | ICAM1 | 5595 | MAPK3 | 54576 | UGT1A8 |
| 841 | CASP8 | 3480 | IGF1R | 5599 | MAPK8 | 54577 | UGT1A7 |
| 847 | CAT | 3552 | IL1A | 5603 | MAPK13 | 54578 | UGT1A6 |
| 885 | CCK | 3553 | IL1B | 5970 | RELA | 54658 | UGT1A1 |
| 1080 | CFTR | 3569 | IL6 | 6288 | SAA1 | 54659 | UGT1A3 |
| 1432 | MAPK14 | 3576 | CXCL8 | 6822 | SULT2A1 | 64240 | ABCG5 |
| 1544 | CYP1A2 | 4129 | MAOB | 6863 | TAC1 | 64241 | ABCG8 |
| 1555 | CYP2B6 | 4313 | MMP2 | 7097 | TLR2 | 94233 | OPN4 |
| 1557 | CYP2C19 | 4843 | NOS2 | 7124 | TNF | | |

AGE-RAGE signaling pathway in diabetic complications, Toll-like receptor signaling pathway, and TNF signaling pathway, etc. (Fig 4A and 4B). In addition, the interactions among 83 potential targets, top 15 BP terms, and top 15 pathways were visualized by a BP-target-pathway network (Fig 5A). Furthermore, the interactions among top 10 hub targets, namely ALB, IL6, AKT1, TP53, TNF, MAPK3, APOE, IL1B, PPARG, and PPARA, were visualized by a PPI network. The network showed 10 target nodes connected by 44 edges with an average degree of 8.8 (Fig 5B).

## Molecular docking between bioactive compounds and hub targets

Molecular docking was performed to validate the interactions between bioactive compounds and hub targets of SHCZF against cholestasis. Seven main compounds of SHCZF, including chrysophanol, emodin, physcion, rhein, aloe-emodin, berberine chloride, and gardenoside, were chosen for molecular docking based on their high contents analyzed by HPLC. Top 10 hub targets, including ALB, IL6, AKT1, TP53, TNF, MAPK3, APOE, IL1B, PPARG, and PPARA, were chosen for molecular docking based on network pharmacology. The results presented that the molecular docking affinity of seven active compounds with top 10 hub target proteins were all less than -5 kcal/mol (S3 Table). The strongest binding activity between active compounds and hub target proteins were exhibited in Fig 6. APOE displayed the best binding affinity with berberine chloride (affinity = -10.5 kcal/mol), physcion (affinity = -10 kcal/mol), chrysophanol (affinity = -9.9 kcal/mol), emodin (affinity = -9.8 kcal/mol), and rhein (affinity = -9.8 kcal/mol) (Fig 6A–6E). AKT1 had a strong affinity with berberine chloride (affinity = -10.4 kcal/mol), chrysophanol (affinity = -9.7 kcal/mol), physcion (affinity = -9.7 kcal/mol), and rhein (affinity = -9.7 kcal/mol) (Fig 6F–6I). TP53 bound to emodin with a binding energy of -9.5 kcal/mol (Fig 6J). According to the molecular docking diagrams, the structures of

**Table 3. 141 bioactive compounds of SHCZF against cholestasis.**

| Compound Name | QED | Compound Name | QED |
|---|---|---|---|
| (-)-Epicatechin-pentaacetate | 0.3317 | Istidina | 0.4207 |
| (+)-trans-Carveol | 0.5719 | jatrorrhizine | 0.7352 |
| (Z,Z)-farnesol | 0.6157 | kaempferol | 0.6372 |
| vanillin | 0.5173 | lauric acid | 0.3925 |
| 2-heptanone | 0.5103 | l-carvone | 0.5247 |
| 3,4,5-trihydroxybenzoic acid | 0.4656 | L-Ile | 0.4718 |
| acetic acid | 0.4199 | limonin | 0.4519 |
| adonitol | 0.3082 | linalool | 0.6172 |
| aloe-emodin | 0.7330 | linolenic acid | 0.3326 |
| alpha-humulene | 0.4851 | L-Limonene | 0.4838 |
| alpha-limonene | 0.4838 | L-Lysin | 0.2814 |
| alpha-linolenic acid | 0.3326 | LPG | 0.3562 |
| angelicin | 0.4354 | Lutein | 0.2035 |
| angelicin | 0.4670 | L-Valin | 0.4120 |
| Apocynin | 0.6736 | L-valine | 0.4266 |
| Auraptene | 0.4124 | MAE | 0.4992 |
| Azole | 0.4642 | menthyl acetate | 0.6510 |
| Baicalin | 0.3617 | Methose | 0.3101 |
| berberine | 0.6633 | Methyl naphthalene | 0.5294 |
| berberine | 0.8245 | methyl palmitate | 0.2468 |
| beta-elemene | 0.5799 | Methyleugenol | 0.6599 |
| beta-sitosterol | 0.4354 | MTL | 0.2704 |
| caffeic acid | 0.4750 | myristic acid | 0.4490 |
| caprylic acid | 0.5818 | naphthalene | 0.5114 |
| capsaicin | 0.5370 | nonanoic acid | 0.5775 |
| chrysin | 0.8206 | obaculactone | 0.4519 |
| cinnamic acid | 0.6504 | Obacunone | 0.4784 |
| cis-Carveol | 0.5719 | o-caffeoylquinic acid | 0.2356 |
| citric acid | 0.4243 | octanoic acid | 0.5818 |
| coumarin | 0.4124 | octanol | 0.5480 |
| crocetin | 0.5030 | oleanolic acid | 0.4460 |
| Cyclopentenone | 0.4228 | oleic acid | 0.2030 |
| DBP | 0.4752 | OYA | 0.3958 |
| DEP | 0.6925 | PAC | 0.6684 |
| DIBP | 0.6761 | paeonol | 0.5478 |
| DLA | 0.4605 | palmatine | 0.6613 |
| d-limonene | 0.4838 | palmitic acid | 0.3653 |
| DTY | 0.5110 | PCR | 0.5390 |
| EIC | 0.2944 | PEA | 0.6259 |
| emodin | 0.6835 | pentadecylic acid | 0.4059 |
| esculetin | 0.3579 | PHA | 0.5664 |
| EUG | 0.6993 | phenylalanine | 0.5664 |
| eugenol | 0.6955 | PHPH | 0.5905 |
| farnesol | 0.6157 | PIT | 0.4834 |
| fructose | 0.3101 | PLO | 0.7502 |
| Fumarine | 0.7258 | poriferast-5-en-3beta-ol | 0.4354 |
| Furol | 0.4792 | Prolinum | 0.3867 |

(*Continued*)

**Table 3.** (Continued)

| Compound Name | QED | Compound Name | QED |
|---|---|---|---|
| gallicacid | 0.4656 | puerarin | 0.4049 |
| genipin | 0.5093 | py | 0.4453 |
| geniposide | 0.2532 | quercetin | 0.5064 |
| geraniol | 0.6172 | rhapontigenin | 0.7399 |
| Germacron | 0.4329 | rhein | 0.7375 |
| GLB | 0.3046 | rottlerin | 0.2140 |
| glutamate | 0.3835 | rutaecarpine | 0.6889 |
| guaiacol | 0.5771 | scoparone | 0.5470 |
| guanidine | 0.2426 | scopoletin | 0.5425 |
| Guasol | 0.5771 | Scopoletol | 0.5425 |
| Gulutamine | 0.3835 | Serotonin | 0.6456 |
| Hemo-sol | 0.4838 | serotonine | 0.6456 |
| Heptadekan | 0.2688 | stearic acid | 0.3017 |
| heptanoic acid | 0.5128 | Stigmasterol | 0.4599 |
| Heptanol | 0.5465 | succinic acid | 0.5303 |
| hexanal | 0.2939 | TDA | 0.4900 |
| hexanoic acid | 0.5687 | tetradecane | 0.3217 |
| histidine | 0.4184 | thymol | 0.6510 |
| Hyacinthin | 0.4290 | trans-2-nonenal | 0.3144 |
| IFP | 0.3920 | tridecanoic acid | 0.4900 |
| IPH | 0.5172 | trihydroxybenzoic acid | 0.4656 |
| isoimperatorin | 0.4856 | UND | 0.4133 |
| isoliquiritigenin | 0.5824 | ursolic acid | 0.4433 |
| isorhamnetin | 0.6678 | | |

emodin bound to sites of ALA-260 and LYS-268, while rhein interacted with LEU-330 in APOE by hydrogen bond (Fig 6D and 6E). Berberine chloride bound to sites of ARG-206 and SER-205 in AKT1, while chrysophanol bound to sites of SER-205, LYS-268, and ASN-53 (Fig 6F and 6G). Both physcion and rhein bound to sites of SER-205 and LYS-268 in AKT1 (Fig 6H and 6I). The structure of emodin bound to the site of ASP-65 in TP53 (Fig 6J).

## Discussion

Cholestasis is clinical condition and pathogenic features caused by the impairment of bile flow, which is closely associated with hepatocyte dysfunction and liver diseases [38]. Previous study indicated that SHCZF had the potential for cholestasis treatment, however, the pharmacological mechanisms remain unclear [6]. Our study found that SHCZF possessed 7 major chemical compounds, including chrysophanol, emodin, physcion, rhein, aloe-emodin, berberine chloride, and gardenoside. According to the network pharmacology analysis, 141 bioactive compounds and 83 potential targets of SHCZF against cholestasis were screened. The corresponding biological functions of potential targets were characterized and presented by Go terms and KEGG pathways. Furthermore, the interactions between 7 major bioactive compounds and top 10 hub target proteins were exhibited by molecular docking.

SHCZF is a Chinese medicine formula, presenting a hepatoprotective effect on intrahepatic cholestasis [6]. There are five herbs in SHCZF, including Dahuang (*Rhei Radix Et Rhizome*), Huangbai (*Phellodendri Chinrnsis Cortex*), Huangzhizi (*Gardeniae Fructus*), Chaihu (*Radix Bupleuri*) and Baizhu (*Atractylodes Macrocephala Koidz.*). Our study identified 7 major

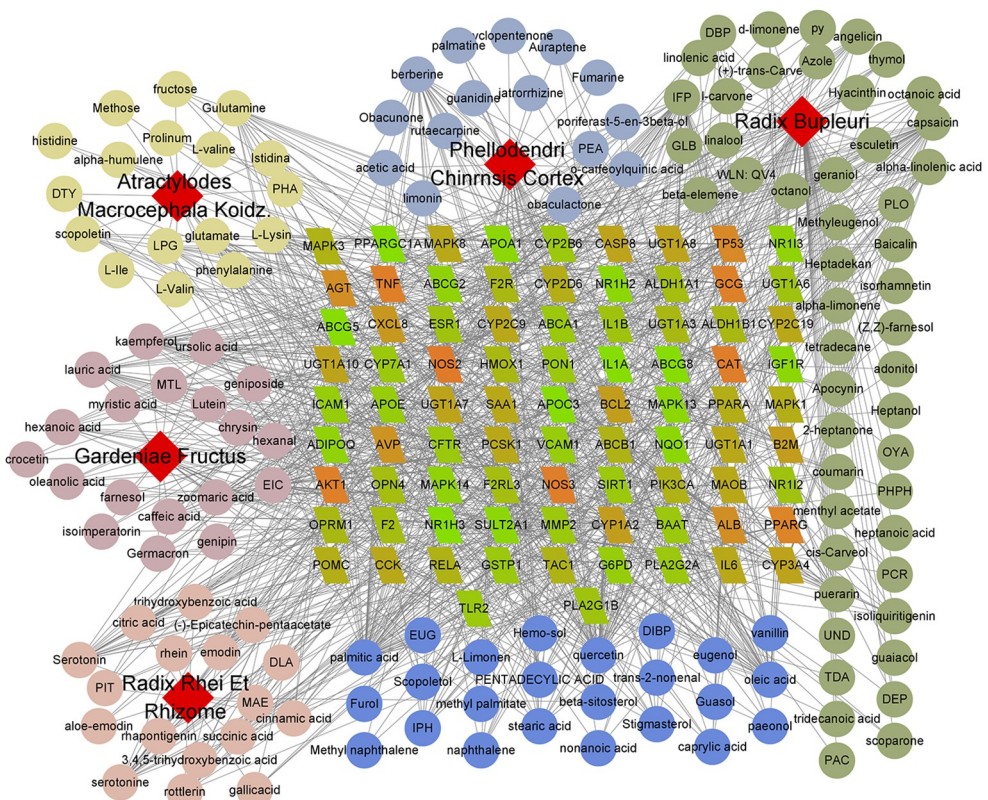

**Fig 2. Compound-target (C-T) network of 141 bioactive compounds and 83 potential targets for SHCZF against cholestasis.** There were 229 nodes in the C-T network, including 5 red diamonds for herbs from SHCZF, 83 orange (higher degree) and green (lower degree) parallelograms for potential targets, and 141 circles for bioactive compounds.

chemical compounds in five herbs of SHCZF, including chrysophanol, emodin, physcion, rhein, aloe-emodin, berberine chloride, and gardenoside. Previous studies indicated that these seven compounds have favorable pharmacological properties including anticancer, hepatoprotective, anti-inflammatory, etc. [39–45]. For instance, emodin can suppress liver injury and bile acids secretion, and exert a protective effect on intrahepatic cholestasis [40]. Physcion is a novel liver protective agent by reprogramming the hepatic circadian clock [41]. Rhein may promote bile acid transport and reduce bile acid accumulation in liver [42]. As a result, we speculate that these seven compounds from SHCZF may exert critical effects for SHCZF against cholestasis.

Network pharmacology are widely applied in elucidating the biological mechanism of traditional Chinese medicine formula by constructing intricate interaction network based on bioactive compounds, targets, and biological functions [46]. According to the network pharmacology analysis, a total of 141 bioactive compounds and 83 potential targets of SHCZF against cholestasis were collected based on public databases. The interactions among 83 targets were presented by a PPI network containing 83 target nodes connected by 1034 edges with an average node degree of 24.9. Besides, the interactions between 141 bioactive compounds and 83 potential targets were visualized by a C-T network. The top 10 hub targets were ALB, IL6, AKT1, TP53, TNF, MAPK3, APOE, IL1B, PPARG, and PPARA. Of note, most of them is associated with the progression of liver diseases [47–51]. For instance, ALB is a protein produced by liver, which is widely used as a marker for liver diseases [47]. IL6 and TNF are inflammatory

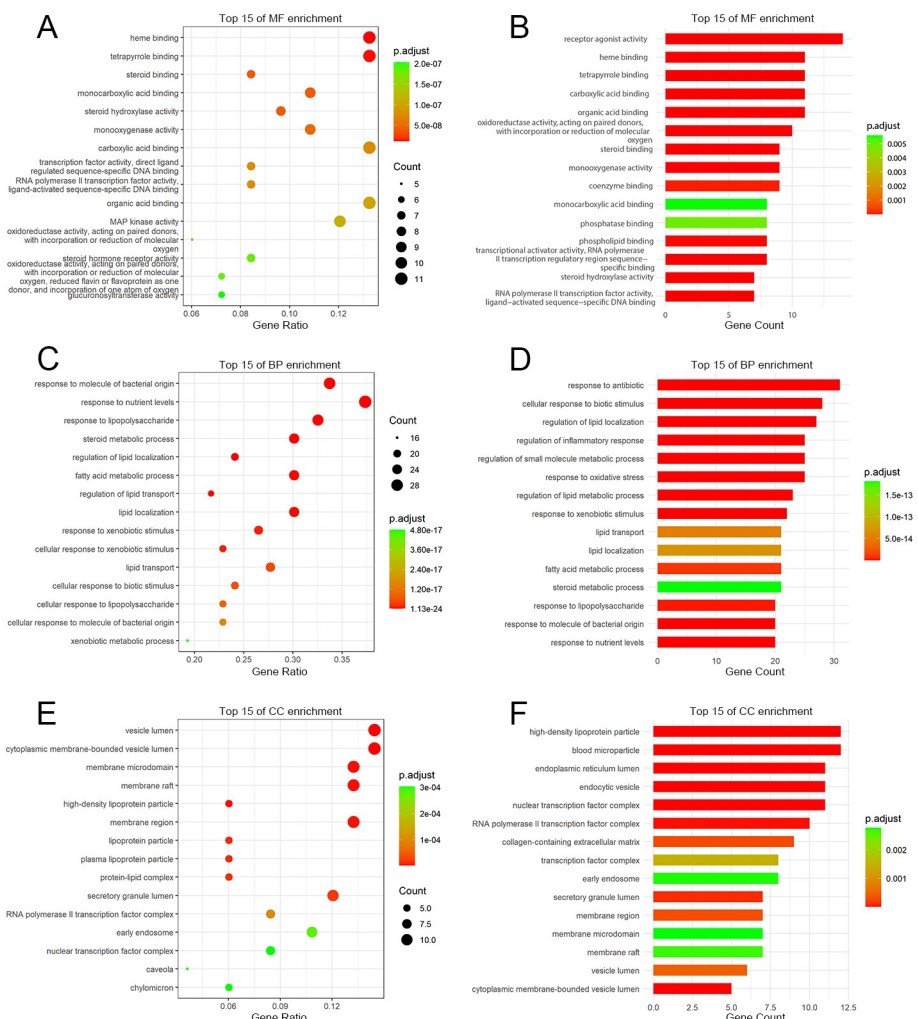

**Fig 3. Gene Ontology (GO) enrichment analysis for 83 potential targets of SHCZF against cholestasis. (A, B)** The bubble chart and histogram of top 15 molecular function (MF) enrichment. **(C, D)** The bubble chart and histogram of top 15 biological process (BP) enrichment. **(E, F)** The bubble chart and histogram of top 15 cellular component (CC) enrichment.

biomarkers for cholestatic liver injury [48]. AKT1 and TP53 are closely related to the regulation of liver cancer progression [50, 51]. These results suggest that these top 10 hub targets may act as essential roles in SHCZF for cholestasis treatment.

In order to further investigate the underlying mechanisms, the biological functions of hub targets were enriched via GO and KEGG analyses. The interactions among 83 potential targets, top 15 related BP terms, and top 15 KEGG pathways were presented by a BP-target-pathway network. Our study showed that these targets were mainly related to the processes of response to molecule of bacterial origin, response to nutrient levels, response to lipopolysaccharide, etc. A previous study also found that patients with cholestasis presented a lack of response to bacterial infections [52]. These results suggested that these targets may be involved in the regulation of SHCZF against cholestasis via moderating these biological processes. In addition, the top 15 KEGG pathways related to hub targets were AGE-RAGE signaling pathway in diabetic complications, fluid shear stress and atherosclerosis, drug metabolism-cytochrome P450, TNF

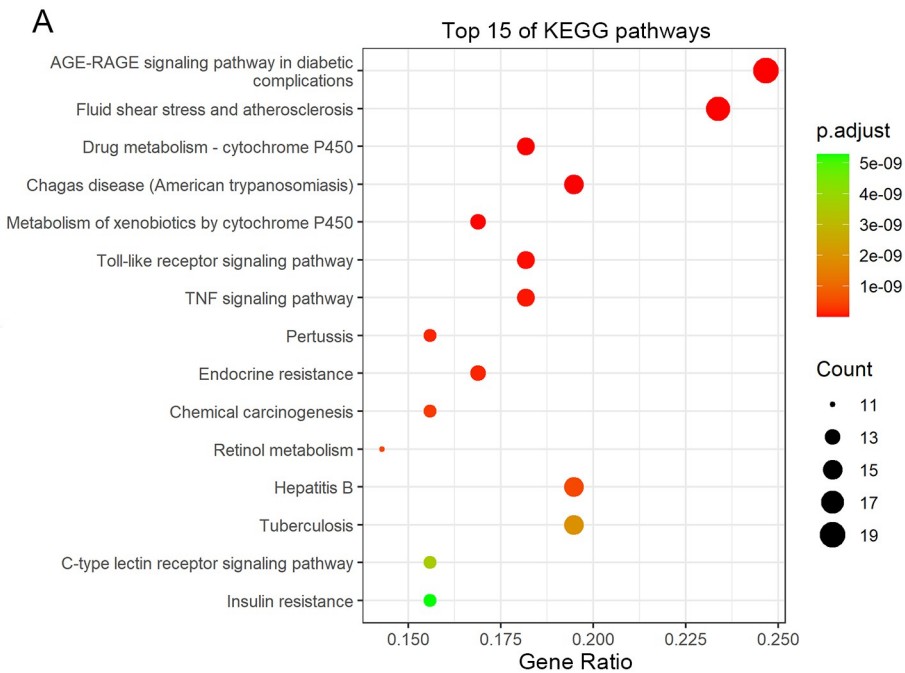

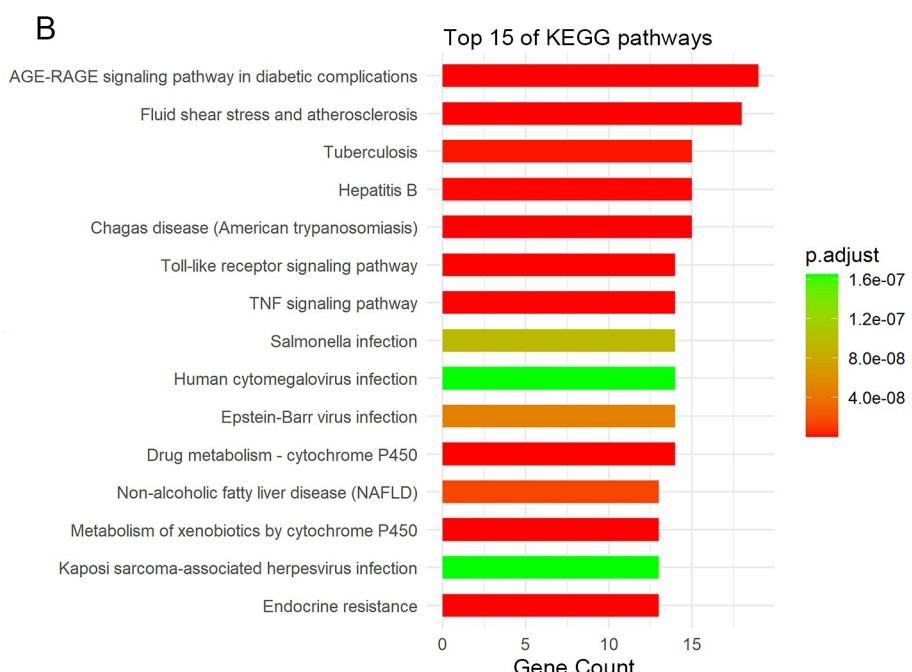

**Fig 4. Kyoto Encyclopedia of Genes and Genomes (KEGG) enrichment analysis for 83 potential targets of SHCZF against cholestasis. (A)** The bubble chart of top 15 KEGG pathways. **(B)** The histogram of top 15 KEGG pathways.

signaling pathway, insulin resistance, etc. According to previous statistic, the pathways of AGE-RAGE signaling pathway in diabetic complications, fluid shear stress and atherosclerosis, and insulin resistance were also enriched in non-alcoholic fatty liver and involved in the

**Table 4. Top 15 KEGG pathways for SHCZF against cholestasis.**

| ID | Pathway | *P*. adjust | Count |
|---|---|---|---|
| hsa04933 | AGE-RAGE signaling pathway in diabetic complications | 7.28E-18 | 19 |
| hsa05418 | Fluid shear stress and atherosclerosis | 5.18E-14 | 18 |
| hsa00982 | Drug metabolism—cytochrome P450 | 2.46E-13 | 14 |
| hsa05142 | Chagas disease (American trypanosomiasis) | 1.37E-12 | 15 |
| hsa00980 | Metabolism of xenobiotics by cytochrome P450 | 1.08E-11 | 13 |
| hsa04620 | Toll-like receptor signaling pathway | 2.58E-11 | 14 |
| hsa04668 | TNF signaling pathway | 6.29E-11 | 14 |
| hsa05133 | Pertussis | 1.40E-10 | 12 |
| hsa01522 | Endocrine resistance | 1.50E-10 | 13 |
| hsa05204 | Chemical carcinogenesis | 3.32E-10 | 12 |
| hsa05161 | Hepatitis B | 4.98E-10 | 15 |
| hsa00830 | Retinol metabolism | 4.98E-10 | 11 |
| hsa05152 | Tuberculosis | 1.93E-09 | 15 |
| hsa04625 | C-type lectin receptor signaling pathway | 3.61E-09 | 12 |
| hsa04931 | Insulin resistance | 5.28E-09 | 12 |

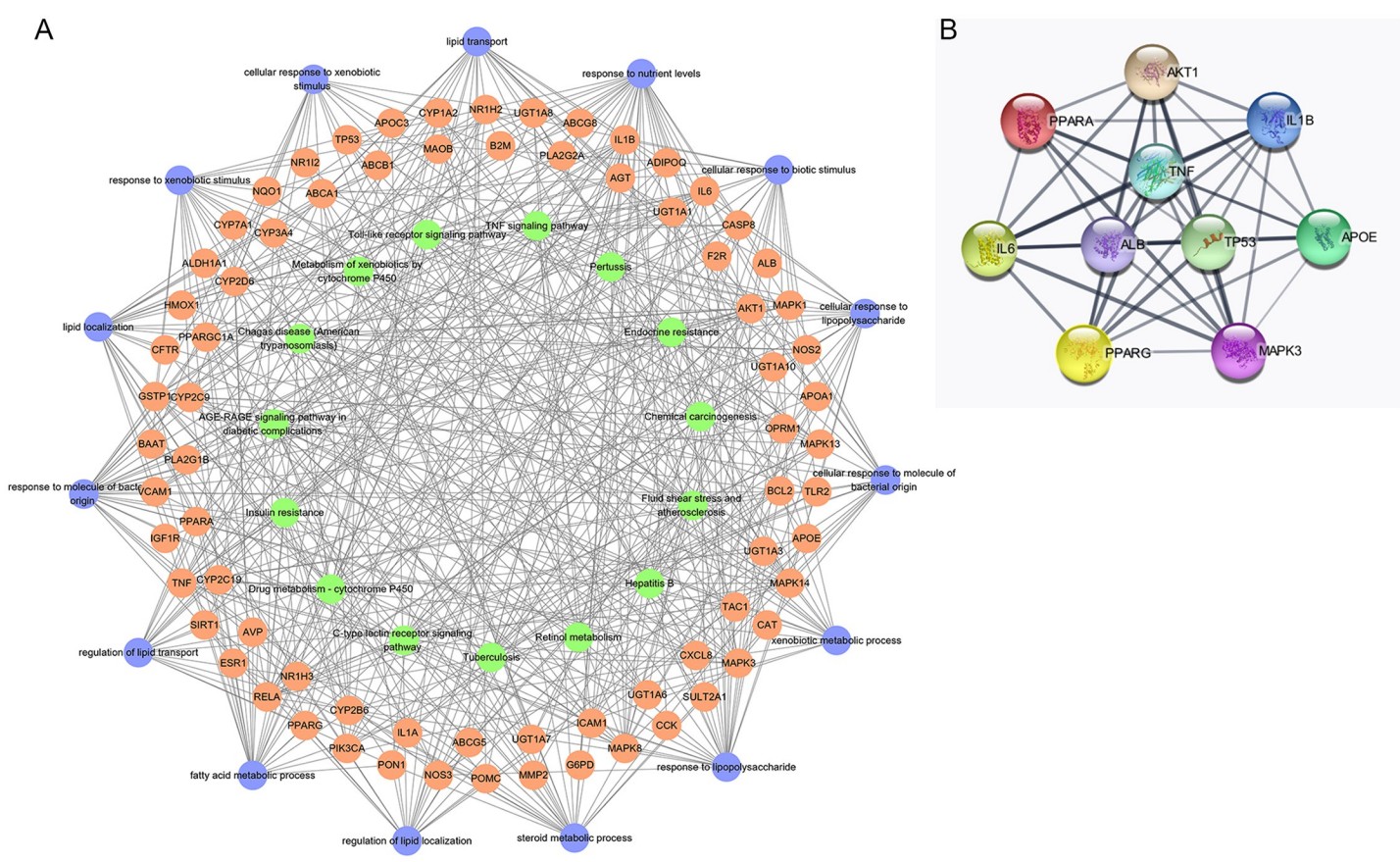

**Fig 5. BP-target-pathway network and PPI network of top 10 hub genes for SHCZF against cholestasis. (A)** The BP-target-pathway network included 83 potential targets (flesh-colored circles), top 15 BP terms (purple circles), and top 15 KEGG pathways (green circles). **(B)** PPI network of top 10 hub targets (ALB, IL6, AKT1, TP53, TNF, MAPK3, APOE, IL1B, PPARG, and PPARA) for SHCZF against cholestasis.

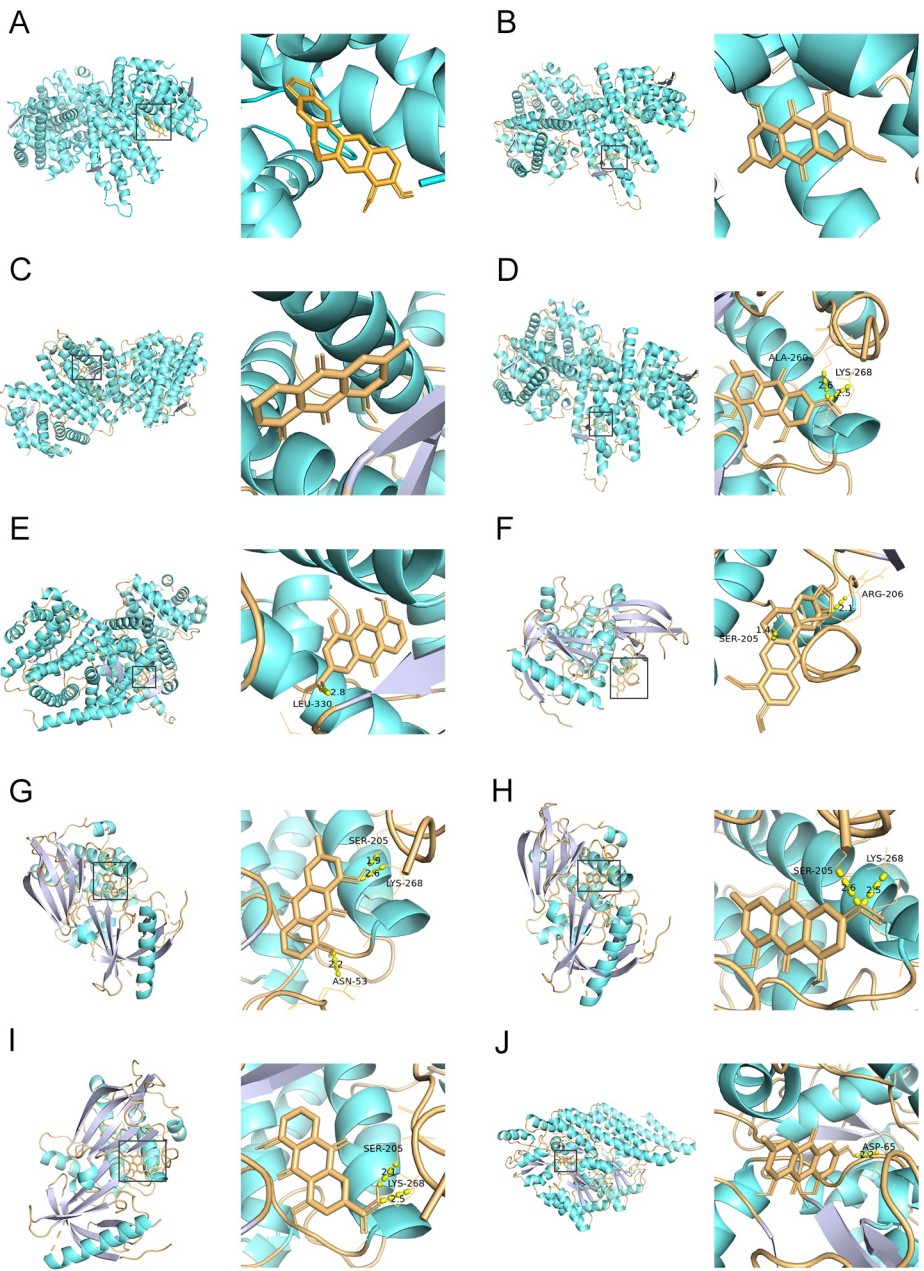

**Fig 6. Molecular docking of SHCZF compounds and hub target proteins. (A-E)** The binding mode of APOE and berberine chloride, physcion, chrysophanol, emodin, and rhein, respectively. **(F-I)** The binding mode of AKT1 and berberine chloride, chrysophanol, physcion, and rhein, respectively. **(J)** The binding mode of TP53 and emodin.

regulation of liver function [53]. Xue et al. [54] found that Da-Huang-Xiao-Shi decoction could upregulate the expression of the metabolic enzyme cytochrome P450 in chronic chole-stasis. Our previous study suggested that TNF signaling pathway may be the important mecha-nism for SHCZF against cholestasis [6]. Overall, the above pathways may be closed relevant to SHCZF against cholestasis.

The binding force of a drug with target proteins is a pivotal index for assessing its mechanistic action on diseases [55]. The binding models between 7 SHCZF compounds and 10 hub target proteins were visualized by molecular docking. The results showed that chrysophanol, physcion, rhein, aloe-emodin, and berberine chloride had a strong affinity with APOE and AKT1. Emodin had a strong affinity with APOE, AKT1, and TP53. The structures of emodin and rhein bound to sites of SER-278 and LEU-330 in APOE, respectively. The structure of berberine chloride bound to sites of ARG-206 and SER-205 in AKT1, while chrysophanol bound to sites of SER-205, LYS-268, and ASN-53. Both physcion and rhein bound to sites of SER-205 and LYS-268 in AKT1. The structure of emodin bound to the site of ASP-65 in TP53. Differences in the binding sites may affect the ability of SHCZF compounds to bind target proteins, thereby exerting regulatory effects on cholestasis.

In conclusion, the interactions of 141 bioactive compounds and 83 potential targets of SHCZF against cholestasis were characterized by network pharmacology analysis. These targets may be closely related to the biological processes of response to molecule of bacterial origin, response to nutrient levels, response to lipopolysaccharide, etc., and involved in the pathways of AGE-RAGE signaling pathway in diabetic complications, fluid shear stress and atherosclerosis, drug metabolism-cytochrome P450, TNF signaling pathway, insulin resistance, etc. Molecular docking validated the binding of 7 active compounds and top 10 hub target proteins. Chrysophanol, physcion, rhein, aloe-emodin, and berberine chloride had a strong affinity with APOE and AKT1, and emodin had a strong affinity with APOE, AKT1, and TP53. This study provides essential clues to further explore the underlying mechanisms of SHCZF against cholestasis. However, *in vivo* or *in vitro* experiments are needed to be performed for validating the mechanisms of SHCZF against cholestasis through moderating above hub targets and pathways.

## Supporting information

**S1 Fig. High Performance Liquid Chromatography (HPLC) chromatograms of 7 major chemical compounds in SHCZF. (A)** Chrysophanol. **(B)** Emodin. **(C)** Physcion. **(D)** Rhein. **(E)** Aloe-emodin. **(F)** Berberine chloride. **(G)** Gardenoside.
(PDF)

**S1 Table. Cholestasis-related targets from public databases.**
(XLSX)

**S2 Table. 162 active compounds and 457 corresponding targets of SHCZF.**
(XLSX)

**S3 Table. Molecular docking of seven bioactive compounds and top 10 targets.**
(DOCX)

## Author Contributions

**Conceptualization:** Binbin Liu, Jiaming Yao.

**Data curation:** Binbin Liu, Jie Zhang, Lu Shao, Jiaming Yao.

**Funding acquisition:** Jiaming Yao.

**Writing – original draft:** Binbin Liu.

**Writing – review & editing:** Jiaming Yao.

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
