## [Decision Letter · Decision Letter 0]

8 Nov 2021

PONE-D-21-30868Network pharmacology analysis and molecular docking to unveil the potential mechanisms of San-Huang-Chai-Zhu formula treating cholestasisPLOS ONE

Dear Dr. Yao,

Thank you for submitting your manuscript to PLOS ONE. After careful consideration, we feel that it has merit but does not fully meet PLOS ONE’s publication criteria as it currently stands. Therefore, we invite you to submit a revised version of the manuscript that addresses the points raised during the review process.  The study has merit.

We look forward to receiving your revised manuscript.

Kind regards,

Gianfranco D. Alpini

Academic Editor

PLOS ONE

Journal Requirements:

“This work was supported by Zhejiang science and technology research fund [No.2014C33238] and Zhejiang science and technology research fund of traditional Chinese medicine [No.2020ZB158].”

Reviewers' comments:

Reviewer's Responses to Questions

**Comments to the Author**

1. Is the manuscript technically sound, and do the data support the conclusions?

Reviewer #1: Yes

Reviewer #2: Yes

2. Has the statistical analysis been performed appropriately and rigorously? 

Reviewer #1: Yes

Reviewer #2: Yes

3. Have the authors made all data underlying the findings in their manuscript fully available?

Reviewer #1: Yes

Reviewer #2: Yes

4. Is the manuscript presented in an intelligible fashion and written in standard English?

Reviewer #1: Yes

Reviewer #2: Yes

5. Review Comments to the Author

Reviewer #1: In this report, Yao et al present a very through and complex bioinformatic analysis of the molecular mechanism of a Chinese medicine formulae, San-Huang-Chai-Zhu formula (SHCZF), against cholestasis. They identified 7 major active chemical compounds of SHCZF by HPLC and revealed their targets via data mining. By comparing with the cholestasis-related genes that retrieved from public databases, 83 over-lapping targets were pinned down as core targets of the SHCZF active compounds against cholestasis. Although this study didn’t provide any rigor and independent confirmation due to the scope of the research, it provided some profound insight of the underlying mechanism of SHCZF against cholestasis, which is highly novel and significant in the larger scheme of developing novel therapeutic strategy against cholestasis. I would suggest the manuscript be accepted in the current form with some minor modifications such as:

• Define the abbreviations at the first use for the less initiated, such as: GO=Gene Ontology; CC=Cellular Component.

• Detailed results description are strongly advised for a better understanding of the analysis. For example, in molecular docking analysis, the authors should provide some more details about the interactions between the active compounds and their target molecules.

Reviewer #2: The manuscript submitted by Liu et al used network pharmacology and molecular docking approach to identify the targets of key ingredients/ compounds found in the formulated Chinese medicine SHCZF. The data and methods presented are clear and detailed. Some minor comments to be addressed before it is considered for publication.

1. Figure 5 and 6 can be part of the same figure.

2. Table S1 presents critical data and should become part of the main manuscript.

3. Figure 7 needs some more annotation/ labelling.Apart from the animo acids, the interacting molecules are not annotated in the figure.

4. Please proof-read for minor drafting issues.

6. PLOS authors have the option to publish the peer review history of their article (what does this mean?). If published, this will include your full peer review and any attached files.

Reviewer #1: **Yes: **Wenjun Zhang

Reviewer #2: No

---

## [Author Response · Author response to Decision Letter 0]

16 Nov 2021

Response to Reviewers

The authors would like to appreciate the editor and reviewers for their precious time and invaluable comments for our work. We have carefully revised our manuscript according to the reviewers’ comments. We believe that the manuscript has been further improved. The corresponding revisions and refinements made in the manuscript are summarized in our response below.

Reviewer #1: In this report, Yao et al present a very through and complex bioinformatic analysis of the molecular mechanism of a Chinese medicine formula, San-Huang-Chai-Zhu formula (SHCZF), against cholestasis. They identified 7 major active chemical compounds of SHCZF by HPLC and revealed their targets via data mining. By comparing with the cholestasis-related genes that retrieved from public databases, 83 over-lapping targets were pinned down as core targets of the SHCZF active compounds against cholestasis. Although this study didn’t provide any rigor and independent confirmation due to the scope of the research, it provided some profound insight of the underlying mechanism of SHCZF against cholestasis, which is highly novel and significant in the larger scheme of developing novel therapeutic strategy against cholestasis. I would suggest the manuscript be accepted in the current form with some minor modifications such as:

Q: Define the abbreviations at the first use for the less initiated, such as: GO=Gene Ontology; CC=Cellular Component.

R: We thank the reviewer for pointing this out. We have checked the full article and defined all abbreviations at the first use in the manuscript.

Q: Detailed results description are strongly advised for a better understanding of the analysis. For example, in molecular docking analysis, the authors should provide some more details about the interactions between the active compounds and their target molecules.

R: Thank you for this professional suggestion. We have added more details in the manuscript about molecular docking, including the binding affinity and sites between active compounds and hub target proteins.

Reviewer #2: The manuscript submitted by Liu et al used network pharmacology and molecular docking approach to identify the targets of key ingredients/ compounds found in the formulated Chinese medicine SHCZF. The data and methods presented are clear and detailed. Some minor comments to be addressed before it is considered for publication.

Q: 1. Figure 5 and 6 can be part of the same figure.

R: We agree the reviewer’s suggestion and have merged Figure 5 and 6 into the same figure (revised Fig. 5). 

Q: 2. Table S1 presents critical data and should become part of the main manuscript.

R: Although we tend to agree the reviewer, the data in the Table S1 are too large to be included in the main manuscript. In addition, the 83 potential targets screened from the data of Table S1 are more important and have been included in the main manuscript. Thus, we provided Table S1 as a supplementary material. 

Q: 3. Figure 7 needs some more annotation/labelling. Apart from the amino acids, the interacting molecules are not annotated in the figure.

R: Thanks for pointing this out. We have added more annotations in Fig. 6 (Fig. 7 was changed to Fig. 6), including interacting molecules by hydrogen bonds. 

Q: 4. Please proof-read for minor drafting issues.

R: We have checked the full article and revised drafting issues with track changes in the manuscript.

---

## [Decision Letter · Decision Letter 1]

10 Feb 2022

Network pharmacology analysis and molecular docking to unveil the potential mechanisms of San-Huang-Chai-Zhu formula treating cholestasis

PONE-D-21-30868R1

Dear Dr. Jiaming Yao,

We’re pleased to inform you that your manuscript has been judged scientifically suitable for publication and will be formally accepted for publication once it meets all outstanding technical requirements.

Kind regards,

Gianfranco D. Alpini

Academic Editor

PLOS ONE

Additional Editor Comments (optional):

Reviewers' comments:

Reviewer's Responses to Questions

**Comments to the Author**

1. If the authors have adequately addressed your comments raised in a previous round of review and you feel that this manuscript is now acceptable for publication, you may indicate that here to bypass the “Comments to the Author” section, enter your conflict of interest statement in the “Confidential to Editor” section, and submit your "Accept" recommendation.

Reviewer #1: All comments have been addressed

2. Is the manuscript technically sound, and do the data support the conclusions?

Reviewer #1: Yes

3. Has the statistical analysis been performed appropriately and rigorously? 

Reviewer #1: Yes

4. Have the authors made all data underlying the findings in their manuscript fully available?

Reviewer #1: Yes

5. Is the manuscript presented in an intelligible fashion and written in standard English?

Reviewer #1: Yes

6. Review Comments to the Author

Reviewer #1: (No Response)

7. PLOS authors have the option to publish the peer review history of their article (what does this mean?). If published, this will include your full peer review and any attached files.

Reviewer #1: **Yes: **Wenjun Zhang

---

## [Editor Report · Acceptance letter]

14 Feb 2022

PONE-D-21-30868R1 

Network pharmacology analysis and molecular docking to unveil the potential mechanisms of San-Huang-Chai-Zhu formula treating cholestasis 

Dear Dr. Yao:

I'm pleased to inform you that your manuscript has been deemed suitable for publication in PLOS ONE. Congratulations! Your manuscript is now with our production department. 

Kind regards, 

on behalf of

Dr. Gianfranco D. Alpini 

Academic Editor

PLOS ONE